# Multi-Fidelity Modeling of Spatio-Temporal Fields

**Sudeepta Mondal, Soumalya Sarkar**

Raytheon Technologies Research Center
411 Silver Lane, East Hartford, CT 06108

## Abstract

Devising emulation frameworks for predicting the spatio-temporal behavior of nonlinear dynamical systems is a challenging problem which has generated significant interest lately. However, scarcity of data often plagues the prevalent approaches, particularly when the cost of procuring spatio-temporal data from the underlying physical processes is high. However, the data sources for a particular spatio-temporal process may present a hierarchy of fidelities with respect to their computational cost/accuracy, such that higher fidelity levels are more accurate (and also more expensive) than the lower fidelity levels. This paper presents a novel multi-fidelity spatio-temporal modeling approach (MF-STM), whereby the lower fidelity data source for a dynamical process is gainfully utilized in increasing the accuracy of predicting the higher fidelity fields. The methodology is based on non-intrusive reduced order modeling using deep convolutional autoencoders, combined with a latent-space evolution framework based on multi-fidelity Gaussian processes. This framework results in probabilistic spatio-temporal predictions for unknown operating conditions of the dynamical system, which provides the end user with quantified levels of uncertainties associated with the data-driven predictions. The framework is validated on an advection-dominated fluid flow process described by the inviscid shallow water equations, which is a well-studied benchmark problem.

Machine learning (ML) frameworks for emulating spatio-temporally varying fields encountered in nonlinear dynamical systems have received a lot of attention in the recent years (Fukami et al. 2021). Data from such systems are often generated by high fidelity (HF) simulations that are multi-physics and multi-scale in nature. Such simulations are often very expensive to perform and thereby incur large computational costs in data generation. Moreover, data generated from such systems often reside in very high dimensional spaces. To this end, there have been advancements in intrusive (Borggaard, Iliescu, and Wang 2011) and non-intrusive (Maulik, Lusch, and Balaprakash 2021) reduced order modeling (ROM) techniques using ML, which have provided promising solutions for constraining the data from such systems in a compressed space, with minimal losses incurred due to data compression. Non-intrusive methods

focus on emulating the spatio-temporal processes by directly learning from data. In problems where there is a lack of understanding of the underlying governing equations for the physical processes which generate the data, non-intrusive methods of ROMs are particularly useful. Specifically, deep learning models such as convolutional neural networks (CNN), long short-term memory networks (LSTM), fully connected neural networks (FCNN) have been extensively used along with more traditional approaches such as proper orthogonal decomposition (POD) and dynamic mode decomposition (DMD) in different application domains. In fact, deep learning models have been widely successful in data-driven emulation tasks due to their higher representation power through arbitrarily complex nonlinear transformations. However, despite the success of such data-driven approaches, the performance of such models tend to be poor when the amount of data is limited. To this end, the state-of-the-art approaches in ROMs and spatio-temporal emulation have rarely catered to data scarce applications.

Data scarcity is a very prevalent problem in practical engineering design scenarios where the simulation models of the underlying physical processes are extremely expensive due to the need for resolving fine spatial and temporal scales. Most advanced machine learning techniques have poor prediction accuracies and generalization guarantees when the amount of training data is limited. To this end, there have been significant developments in the last few years in probabilistic modeling approaches to develop multi-fidelity modeling (MFM) frameworks for addressing the issue of data scarcity in predictive modeling (Giselle Fernández-Godino et al. 2019; Sarkar et al. 2019; Mondal, Joly, and Sarkar 2019). MFM caters to a class of surrogate modeling approaches, whereby a statistical model is developed by learning the correlation and discrepancy among different levels of fidelities of the prevalent models for a physical process. The approach is applicable to scenarios where there exists a hierarchy of the computational models/data sources with respect to their accuracies and computational expenses, such that the higher fidelity levels are more accurate, but they are also more expensive. In such a setting, MFM strategies aim to leverage more data from lower fidelity sources, and less data from higher fidelity sources to formulate an integrated modeling framework which can achieve high accuracy in modeling the input-output relationships of the HF sources.

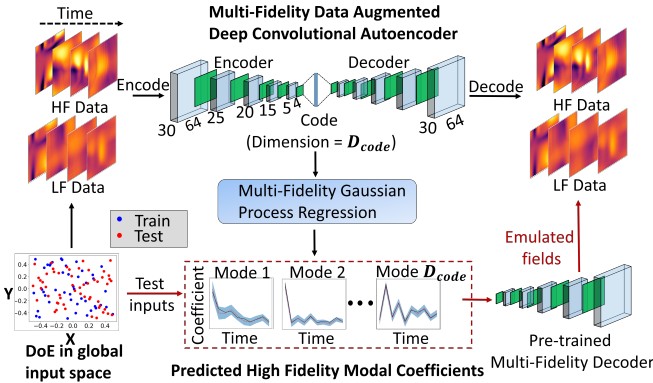

Figure 1: Schematic overview of the proposed multi-fidelity spatio-temporal modeling framework (MF-STM).

In this paper, we propose a multi-fidelity spatio-temporal modeling approach (MF-STM) which leverages low fidelity (LF) data for a non-linear dynamical system along with limited HF data, to achieve better emulation accuracy than the state-of-the-art which solely leverages HF data. In particular, we propose :

- A novel multi-fidelity data augmented convolutional autoencoder (MF-CAE) framework to constrain the representation of low and HF data under a common bottleneck representation, and
- A novel approach of probabilistic latent space interpolation by leveraging state-of-the-art developments in multi-fidelity Gaussian processes (MFGP), which automatically provides quantified estimates of prediction uncertainties.

The key area of novelty in our approach is in its extension of the state-of-the-art spatio-temporal modeling frameworks to enable the leveraging of prevalent multi-fidelity information for better and more efficient predictive modeling. In most applications, data from lower fidelity sources is rejected solely because they do not provide accurate estimates of the physical processes due to the lack of the necessary resolution of temporal/spatial scales and/or relevant physics to do so. Through this work, we claim that if there exists some degree of measurable correlation in the different fidelities of the data sources (which our framework automatically learns), multi-fidelity adaptation can significantly improve the prediction accuracy and reduce the uncertainty of the data-driven predictions under HF data limitations.

## MF-STM Framework

The emulation framework comprises three stages :

1. **Reduced order modeling** : The model order reduction methodology involves a deep MF-CAE framework which learns an efficient coded representation from the multi-fidelity spatio-temporal field data, denoted by $\Psi_{F_j}(\mathbf{s}, t)$ for $j \in 1, 2, \cdots K$, where $j$ is the fidelity index, with $j = 1$ denoting the highest fidelity level and $j = K$ denoting the lowest fidelity level. The spatial

and temporal domains for the data, denoted by $\mathbf{s} = \{s_1, s_2, \cdots s_N\}$ and $t = \{t_1, t_2, \cdots t_N\}$ are assumed to be finite. Moreover, the spatial and temporal domains are assumed to be the same for all the fidelity levels. The data is generated by performing a design of experiments (DoE) in the global input parameter space ($\mathbf{X}_{in}$) for the underlying process. $\Psi_{F_j}$ belongs to a high dimensional space $\mathbb{R}^{D_{in}} \forall j \in 1, 2, \cdots K$. An encoder model ($\mathcal{F}_{enc} : \mathbb{R}^{D_{in}} \to \mathbb{R}^{D_{code}}$) and a decoder model ($\mathcal{F}_{dec} : \mathbb{R}^{D_{code}} \to \mathbb{R}^{D_{in}}$) are the outcomes from this stage, which are separately used in the following two stages. $D_{code}$ represents the dimension of the coding layer, with $D_{code} << D_{in}$. $\mathcal{F}_{enc}$ learns to represent the multi-fidelity data in a reduced dimensional space to provide the coded representations denoted by $\mathbf{z}$, and $\mathcal{F}_{dec}$ learns to reconstruct the high dimensional field data from the coded representations. The reconstruction is snapshot-to-snapshot, hence for each snapshot, the coded output $\mathbf{z}$ denotes the spatial modal coefficients. A temporal sequence of field data $\Psi_{F_j}(\mathbf{s}, t)$ is coded into temporal evolution of the modal coefficients denoted by $\mathbf{z}(t)$.

2. **Latent space interpolation**: Once the coded representations are learned in Stage 1, the temporal evolution of the latent space modes is learned as a function of the input parameters. State-of-the-art MFGP techniques are used for this stage to leverage the correlation among the multiple fidelity levels with respect to their corresponding modal evolution in the latent space. The outcome from this stage is a probabilistic latent space interpolation model denoted by $\mathcal{R} : (\mathbf{X}_{in}, t) \mapsto \mathbf{z}$. This provides the mapping from $\mathbf{X}_{in}$ to $\mathbf{z}(t)$.

3. **Field data emulation**: For an unknown test input $\mathbf{X}_{test} \in \mathbf{X}_{in}$, Stage 2 provides estimates of $\mathbf{z}(t)$. These predicted modes are used as inputs to the pre-trained $\mathcal{F}_{dec}$ from Stage 1, which generate the spatio-temporal flowfields $\Psi_{F_j}(\mathbf{s}, t)$ for the unknown test input condition, thus completing the emulation framework. Figure 1 shows a schematic of the MF-STM framework, as discussed above.

## Results and Discussions

In this work, the MF-STM approach has been demonstrated in a two fidelity setting with fine and coarse grid solutions for the inviscid shallow water equations. Shallow water equations (Equations 1 - 3) belong to a prototypical system of equations for geophysical flows, and the solutions to these equations have been used extensively by researchers for validation and benchmarking of spatio-temporal modeling frameworks.

$$\frac{\partial(\rho\eta)}{\partial t} + \frac{\partial(\rho\eta v_x)}{\partial x} + \frac{\partial(\rho\eta v_y)}{\partial y} = 0 \tag{1}$$

$$\frac{\partial(\rho\eta v_x)}{\partial t} + \frac{\partial(\rho\eta v_x^2 + \frac{1}{2}\rho g\eta^2)}{\partial x} + \frac{\partial(\rho\eta v_x v_y)}{\partial y} = 0 \tag{2}$$

$$\frac{\partial(\rho\eta v_y)}{\partial t} + \frac{\partial(\rho\eta v_x v_y)}{\partial x} + \frac{\partial(\rho\eta v_y^2 + \frac{1}{2}\rho g\eta^2)}{\partial y} = 0 \tag{3}$$

In the above equations, $\eta$ denotes the total fluid column height, $(v_x, v_y)$ denotes the fluid's horizontal flow velocity components averaged across the vertical fluid column, $g$ is the acceleration due to gravity and $\rho$ is the fluid density which is set to 1.0. The equations are subject to initial conditions given by:

$$\rho\eta(x, y, t = 0) = 1 + e^{-\left(\frac{(x-\bar{x})^2}{2(5\times10^4)^2} + \frac{(y-\bar{y})^2}{2(5\times10^4)^2}\right)} \quad (4)$$

$$\rho\eta v_x(x, y, t = 0) = 0 \quad (5)$$

$$\rho\eta v_y(x, y, t = 0) = 0 \quad (6)$$

Here $(x, y)$ and $t$ are the spatial coordinates of the two-dimensional solution domain, and the temporal variable respectively. $\bar{x}, \bar{y}$ denotes our global input parameter space $\mathbf{X}_{in}$, with $-0.5 \leq \bar{x}, \bar{y} \leq 0.5$. The 2-D solution domain is a square with periodic boundary conditions, and in this work we perform emulation only for the $\rho\eta$ field, so all the field data presented in this paper correspond to $\rho\eta$. Further details regarding the shallow water equations can be found in (Maulik, Lusch, and Balaprakash 2021). The system of equations is solved from $t = 0$ to $t = 0.5$ with a time-step of 0.001 on a square 2-D grid with 64 collocation points for the HF data source, and 32 collocation points for the LF data source. Thus the HF solutions are in a $64 \times 64$ gridded data format, and the LF solutions are in a coarser $32 \times 32$ grid. A design of experiments (DoE) using Latin Hypercube sampling is performed in $\mathbf{X}_{in}$, and 10 evenly spaced snapshots in time are collected from the simulations for each input condition. Figure 2 shows an example of the discrepancies between the HF and LF simulations for an input condition $(\bar{x}, \bar{y}) = (-0.385, -0.015)$. It can be seen that at $t = 0.05$ the LF solution lacks the sharp gradients in spatial features, although there is some noticeable correlation in the spatial distribution of regions of high and low values of $\rho\eta$. The LF solution at $t = 0.5$ lacks the fine-grained features captured in the corresponding HF simulation. The next subsections will focus on how the LF data can be leveraged to not only improve the emulation accuracy of data-driven models using solely the HF data, but also reduce the HF data requirement in achieving comparable accuracy. In the following subsections, several training conditions are considered with different combinations of HF and LF training data, which are described later. A set-aside test set of 55 input locations (comprising 550 snapshots of HF data) in $\mathbf{X}_{in}$ is used for testing the performance of the MF-STM framework for each of the training combinations.

## Reconstruction performance

A deep MF-CAE is constructed for compressing $\Psi_{F_j}$ in a latent space with $D_{code} = 20$. $\mathcal{F}_{enc}$ has 5 convolutional layers, with 30, 25, 20, 15 and 5 filters, respectively, having a $3 \times 3$ kernel size. To reduce the dimension of the output after each convolution step, max-pooling of window-size $2 \times 2$ is employed. $\mathcal{F}_{dec}$ is symmetric to $\mathcal{F}_{enc}$ with respect to its convolutional and up-sampling operations. The MF-CAE architecture is depicted in Figure 1. The data augmentation for the MF-CAE involves training a single CAE

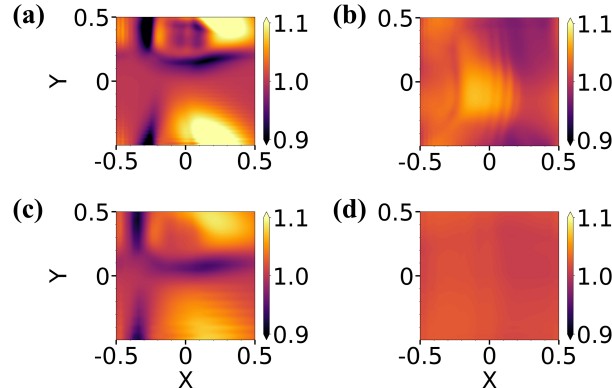

Figure 2: HF field at (a) $t = 0.05$ and (b) $t = 0.5$; LF field at (c) $t = 0.05$ and (d) $t = 0.5$ for $(\bar{x}, \bar{y}) = (-0.385, -0.015)$.

architecture for all the HF and LF data combined. To facilitate this, $\Psi_{F_2}$ is up-sampled using bilinear interpolation to match the dimensions of $\Psi_{F_1}$. Hence, the MF-CAE is trained on $64 \times 64$ dimensional snapshots from both the HF and LF simulations. The effect of augmenting the CAE with LF data is compared with the cases when the CAE is trained with single fidelity (i.e. solely the HF) data, and the latter is termed as HF-CAE. Different training data combinations are considered for HF-CAE, with the number of high fidelity training points in $\mathbf{X}_{in}$, denoted by $N_H$, selected to be 45, 70 and 95, each being a unique Latin Hypercube DoE. All the MF-CAEs have the same 45 HF input training points as the HF-CAE with $N_H = 45$. The metric selected for evaluating the reconstruction performance on the 55 test locations is an average (over 550 snapshots) of the normalized $L_2$ error ($\varepsilon_R$), where $\varepsilon_R = \frac{||\Psi_{F_1} - \tilde{\Psi}_{F_1}||}{||\Psi_{F_j}||}$, with $\tilde{\Psi}_{F_1}$ denoting the CAE reconstruction of $\Psi_{F_1}$. Table 1 compares $\varepsilon_R$ for the different CAE training combinations. It can be observed from the table that addition of LF points in the MF-CAE architecture results in progressively improved reconstruction performance, with around 25% reduction in $\varepsilon_R$ with MF-CAE ($N_H = 45$, $N_L = 95$), as compared to its HF-CAE counterpart with $N_H = 45$. Moreover, it is also seen that MF-CAE($N_H = 45$, $N_L = 70$) and MF-CAE($N_H = 45$, $N_L = 95$) have similar $\varepsilon_R$ as compared to HF-CAE ($N_H = 70$) and HF-CAE ($N_H = 95$), respectively. This indicates that it is possible to achieve comparable reconstruction performance to higher HF data-based CAE models, by leveraging LF data.

| HF-CAE ($N_H = 45$) | HF-CAE ($N_H = 70$) | HF-CAE ($N_H = 95$) |
|---|---|---|
| 0.0207 | 0.0171 | 0.0150 |

| MF-CAE ($N_H = 45$, $N_L = 45$) | MF-CAE ($N_H = 45$, $N_L = 70$) | MF-CAE ($N_H = 45$, $N_L = 95$) |
|---|---|---|
| 0.0187 | 0.0178 | 0.0154 |

Table 1: $\varepsilon_R$ for the different CAE training combinations.

## Regression and Emulation

Given the training inputs in $\{\mathbf{X}_{in} \times t\}$, the $\mathcal{F}_{enc}$ of MF-CAE provides the latent space ($\mathbf{z}$) representations for both the HF and LF fields. This mapping to the latent space is learnt by the MFGP regressor $\mathcal{R}$, which is accomplished using the non-linear autoregressive Gaussian process (NARGP) approach proposed by Perdikaris et al (Perdikaris et al. 2017). It is also assumed that the training inputs have the nested structure of $\mathcal{D}_{F_1} \subseteq \mathcal{D}_{F_2}$, i.e. the HF training input points are a subset of the LF training input points (Kennedy and O'Hagan 2000). NARGP is a modification of the linear autoregressive MF modeling approach originally proposed by Kennedy and O'Hagan (Kennedy and O'Hagan 2000), in order to incorporate non-linear inter-fidelity correlations. The NARGP formulation for our regression problem can be represented as $\mathbf{z}_1(\bar{x}, \bar{y}, t) = g((\bar{x}, \bar{y}, t), \mathbf{z}_{*2}(\bar{x}, \bar{y}, t))$, where $\mathbf{z}_{*2}(\cdot)$ is the GP posterior of the LF level, and $g(\cdot)$ is also a GP. More details of the mathematical formulation of NARGP can be found in Perdikaris et al. (2017).

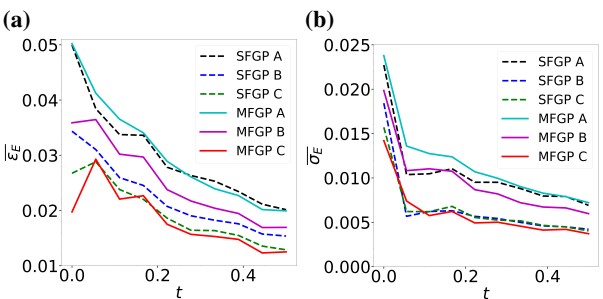

Figure 3: (a) Emulation error $\bar{\varepsilon}_E$ and (b) Emulation uncertainty $\bar{\sigma}_E$ as a function of time, using the six different emulation approaches. Metrics averaged over 55 test samples.

The effect of multi-fidelity modeling in the regression performance has been analyzed using MFGP, and compared with the single fidelity GP (SFGP) counterpart where only the HF data is used both in the ROM and the regression stages. Using SFGP for flowfield emulation combined with ROM has been used by researchers earlier in this context (Maulik et al. 2021). The six different models trained for regression are: SFGP A ($N_H = 45$), SFGP B ($N_H = 70$), SFGP C ($N_H = 95$), MFGP A($N_H = 45$, $N_L = 45$), MFGP B($N_H = 45$, $N_L = 70$) and MFGP C($N_H = 45$, $N_L = 95$). All the MFGPs have the same 45 HF training points as the SFGP with $N_H = 45$. The metrics compared are the normalized $L_2$ error of emulation ($\varepsilon_E$), which denotes the error between the true HF field and the decoded field output from the posterior GP mean prediction of $\mathbf{z}$ for a test input condition, and the standard deviation ($\sigma_E$) of the emulated predictions. These two metrics have been averaged over all the 55 test samples and plotted in Figure 3. It can be seen from Figure 3 that adding LF points progressively reduces the error and the uncertainty of emulation, keeping the same number of HF points. The maximum improvement is obtained for MFGP C when compared with SFGP A, where an improved performance is seen with respect to both the metrics. Moreover, it is interesting to note that the emulation

error of MFGP C is lower than all the SFGPs, with SFGP C being closest to MFGP C. Thus, with 45 HF points and 95 LF points, the multi-fidelity framework could achieve a better emulation performance ($\sim$38% reduction in sample averaged $\varepsilon_E$ and $\sim$42% reduction in sample averaged $\sigma_E$) than a single high fidelity counterpart with 45 HF points. This indicates the effectiveness of incorporating LF data along with the HF information through a multi-fidelity framework, particularly when the LF data is inexpensive to obtain. Figure 4 shows a qualitative comparison among the true HF field and the mean predicted fields from SFGP A and MFGP C for $t = 0.05$ and $t = 0.16$ for a test case with $(\bar{x}, \bar{y}) = (0.445, 0.215)$. It is observed that with the same 45 HF points used in the ROM and regression models, the MFGP C is able to represent the true field more accurately than the single fidelity counterpart, with better overall emulation of the flow features.

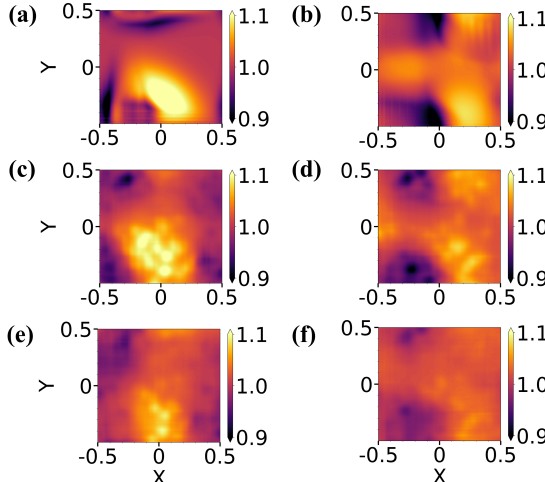

Figure 4: True HF fields at (a) $t = 0.05$ and (b) $t = 0.16$, MFGP C predicted mean HF field at (c) $t = 0.05$ and (d) $t = 0.16$, SFGP A predicted mean HF field at (e) $t = 0.05$ and (f) $t = 0.16$, for a test case with $(\bar{x}, \bar{y}) = (0.445, 0.215)$.

## Conclusions

This paper presents a novel multi-fidelity modeling approach for spatio-temporal emulation, named as MF-STM, that captures the multi-fidelity correlation present in spatio-temporal data. The model has been tested on a prototypical system of equations simulating geophysical flows, namely the inviscid shallow water equations. Augmenting HF data with a relatively inaccurate coarse-grid LF data has resulted in $\sim$25% reduction in reconstruction error for the ROM. Moreover, the MF-STM approach has resulted in $\sim$38% reduction in emulation error and $\sim$42% reduction in prediction uncertainty for the spatio-temporal fields over a held-out test set. The methodology provides a promising approach of making use of the otherwise rejected less accurate (but less expensive) LF data for data-driven emulation, particularly in data-limited scenarios. In situations where the cost difference between the HF and LF models is significant, this framework has the potential of reducing the computational cost in developing accurate and robust surrogate models.

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

# Acknowledgments

This material is based upon work supported by the U.S. Department of Energy's Office of Energy Efficiency and Renewable Energy (EERE) under the Advanced Manufacturing Office, Award Number DE-EE0009398. This report was prepared as an account of work sponsored by an agency of the United States Government. Neither the United States Government nor any agency thereof, nor any of their employees, makes any warranty, express or implied, or assumes any legal liability or responsibility for the accuracy, completeness, or usefulness of any information, apparatus, product, or process disclosed, or represents that its use would not infringe privately owned rights. Reference herein to any specific commercial product, process, or service by trade name, trademark, manufacturer, or otherwise does not necessarily constitute or imply its endorsement, recommendation, or favoring by the United States Government or any agency thereof. The views and opinions of authors expressed herein do not necessarily state or reflect those of the United States Government or any agency thereof. The authors also acknowledge Amit Surana, John J. Gangloff, Wenping Zhao and Lei Xing of Raytheon Technologies Research Center for their valuable comments in shaping this work.

