# OpenReview forum: "Multi-Fidelity Modeling of Spatio-Temporal Fields"
_AAAI.org/2022/Workshop/ADAM — AAAI 2022 Workshop ADAM_

### Official Review · Reviewer_c7av · 2021-11-29
**Well-written paper. Interesting study of multi-fidelity spatio-temporal field prediction using deep learning model with promising preliminary results. Recommend it to be accepted.**

**Rating:** 9
**Confidence:** 5

**Review:**

This paper aims to overcome the bottleneck of data scarcity in predicting the Spatio-temporal field of non-linear dynamic systems when using data-driven models. By combining a series of designed low- and high-fidelity datasets, the proposed deep learning model (MF-CAE) generated promising results compared to single-fidelity CAE. The single-fidelity and multi-fidelity Gaussian Processes (in Stage 2) were also compared to demonstrate the efficacy of the multi-fidelity framework. The framework is clearly described with sound preliminary results.

Pros: Well-written paper. Clear framework and methodology. Promising results demonstrate the efficacy of the framework in solving the data scarcity when high-fidelity data is expensive to obtain.

Cons: Not clear how costly it is to obtain high-fidelity data. When using multi-fidelity data, I'm curious about what the best combinations will be. That means there could be a study to show how the different resolution hierarchies (how low and how high the fidelity should be?) impact final prediction results.

---

### Official Review · Reviewer_LZ9M · 2021-11-30
**Spatiotemporal field prediction using a hybrid ML framework**

**Rating:** 7
**Confidence:** 4

**Review:**

This paper proposes a general purpose ML framework for prediction of spatiotemporal fields. The paper is well-motivated and well-written. However, connecting the work with the central theme of the workshop (AI for design) more strongly would be more useful. The framework has two parts, the first part is an autoencoder that is trained with multi-fidelity data, the second part is a GP that maps the global input parametric condition to the embedding space of the autoencoder. Hence, during testing a parametric input can be converted to a code (using the trained GP) and then to the spatiotemporal field (using the trained decoder). Although some discussion is provided in the results section, it is not entirely clear from the framework description how the GP focuses on predicting modal coefficients that correspond to HF data. The overall framework in figure 1 can also be improved to distinguish between the training and testing stages. Ablation studies in the results section show a lot of promise in terms of leveraging multi-fidelity data as opposed to only expensive HF data.